

# Minimum dissipation of potential energy by groundwater outflow results in a simple linear catchment reservoir

Axel Kleidon[1] and Hubert H G Savenije[2]

[1]Biospheric Theory and Modelling Group, Max-Planck-Institut für Biogeochemie, Jena, Germany
[2]TU Delft, Delft, The Netherlands

*Correspondence to:* Axel Kleidon (akleidon@bgc-jena.mpg.de)

**Abstract.** Streamflow recessions of catchments during periods of no recharge can often be reproduced by a simple, linear reservoir despite the complexity of the catchments. Here we show that such a simple linear behaviour can result from the assumption that groundwater drains from smaller units within the catchment into the stream in such a way that the potential energy of groundwater of the whole catchment is dissipated at the minimum possible rate. To do so, we consider the mass balances of groundwater of two connected sub-catchments that form a hypothetical catchment and consider the depletion of potential energy as groundwater drains into the channel network. We show analytically that the catchment-level depletion of groundwater potential energy has a minimum with respect to a groundwater flux that connects the sub-catchments. The catchment-level minimisation results in equal groundwater levels in the sub-catchments with respect to their channels, which then results in a simple, linear reservoir model for the whole catchment. We then discuss the requirements for such a minimum dissipation state to exist and propose possible mechanisms by which groundwater flow can organise and evolve to such a state. We conclude that the simple, linear response in streamflow recession can be interpreted as the outcome of groundwater flow within the catchment organised to dissipate potential energy at the minimum possible rate. Hence, it would seem that energetic considerations provide an important, additional constraint in the dynamics of water flow networks within catchments that potentially reduces the problem of equifinality in hydrology.

## 1 Introduction

As is commonly known to hydrologists, the tail of a runoff hydrograph at the outfall of a natural catchment can often be very well described by an exponential depletion of the flow (Tallaksen, 1995; Wittenberg and Sivapalan, 1999; Hrachowitz et al., 2013). If we use a logarithmic scale on the vertical axis of a hydrograph (see Figure 1), we can clearly see straight lines during periods of low flow, all having the same inclination. The slope of the line is the time scale of the process, which corresponds with the average residence time of the water within the active groundwater stock draining towards the outfall. If during a period of low flow there is no recharge to the groundwater, and if there is no withdrawal of groundwater by deep rooting vegetation, leakage to neighbouring catchments, or human withdrawal, then the water balance of the groundwater reservoir is simply given by:

$$\frac{dS}{dt} = -Q \qquad (1)$$

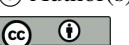


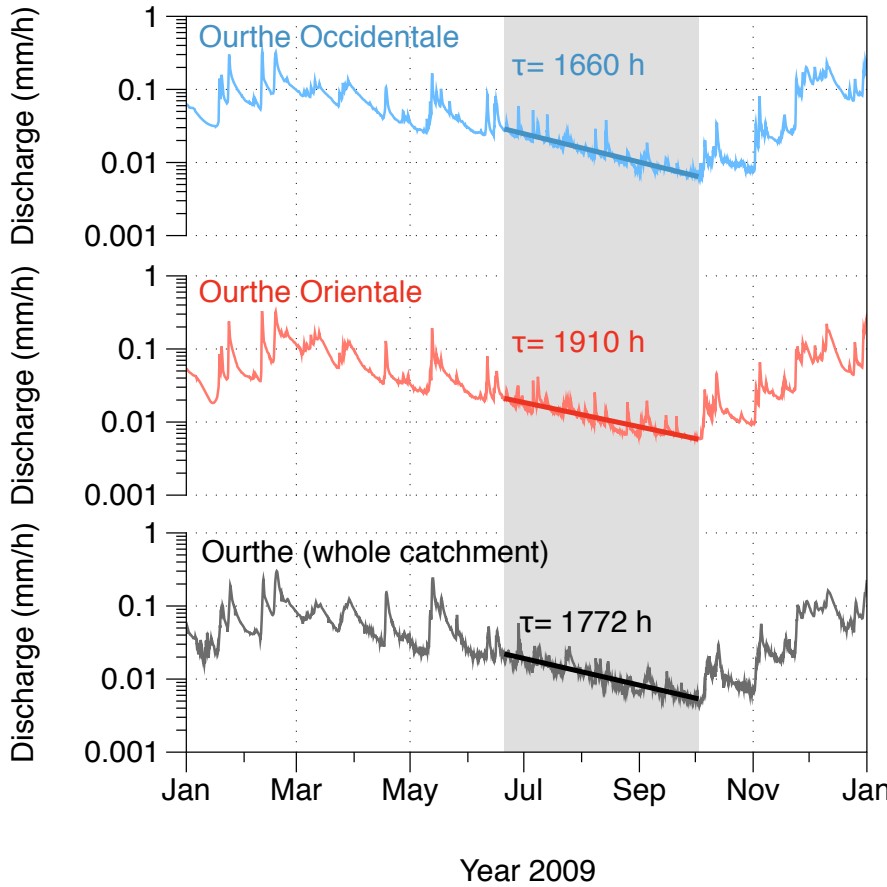

**Figure 1.** Stream hydrograph of the two branches of the Ourthe river sub-catchments (top, middle) as well as of the whole catchment (bottom). The lines represent a best fit during the time period shaded in grey, with the associated time scale of discharge $\tau$ marked in hours.

where $S$ is the active storage of groundwater with a head higher than the level of the water in the stream (measured along the streamline of the groundwater) and $Q$ is the seepage discharge towards the stream, which (assuming minimum storage variation in the stream) is equal to the discharge at the outfall of the catchment. In using this definition of $S$ we limit ourselves to the active part of the groundwater and do not consider the passive storage of groundwater, which makes no contribution to the

5    discharge of the stream at the outfall of the catchment. The exponential recession curve that we often observe in nature follows from the combination of the water balance with a linear reservoir where the discharge of the reservoir is directly proportional to the amount of active storage in the reservoir:

$$Q = \frac{S}{\tau} \tag{2}$$

where $\tau$ is a characteristic time scale of the outflow process.




Combination of Eq. (1) and Eq. (2) leads to the simple exponential recession curve:

$$Q = Q_0 \exp(-t/\tau) \tag{3}$$

where $Q_0$ is the discharge at $t = 0$.

The linear reservoir, introduced by Maillet (1905), is widely used to describe flow recession and can be easily used in water

resources management to predict stream flow if no recharge is expected to occur, or to estimate the amount of active storage still available for water use until the onset of the next rainy season.

Although the linear reservoir assumption is often contested (Wittenberg, 2003; Moore, 1997; Chapman, 1999), particularly by authors who look at processes at smaller scales (hillslopes or small watersheds underlain by an impermeable foundation), or at recessions that are still influenced by recharge from the unsaturated zone, by leakage, or by groundwater subtraction, the

general occurrence of the linear recession is widely reported to exist in natural catchments (e.g. Fenicia et al., 2006).

The big question remains why this is so. Why does such a simple equation pertain, while hydrologists and geologists will agree that the subsurface is extremely heterogeneous, consisting of often complex configurations of different geological strata, networks of fractures in hard rock or conglomerates, separated by isolated pockets and perched aquifers, and connected by a complex network of preferential flow channels through pockets with much lower permeability. But even if the subsurface were a

homogeneous porous medium, within which the groundwater would flow according to Darcy's law with a filter velocity directly proportional to the gradient, even then would such a system only lead to exponential recession under very specific drainage basin configurations (see e.g. Troch et al., 2003). It remains a riddle why complex and heterogeneous systems often demonstrate surprisingly simple behavior (Savenije, 2001). In an attempt to explain this apparently simple behavior, Fenicia et al. (2006) hypothesized that the process of self-organization forming river basin landscapes and drainage networks (Rodriguez-Iturbe and

Rinaldo, 1997) is mirrored underground and that the groundwater drainage system co-evolved with the geological formation and erosion processes, leading to efficient dissipation of the available head, that is, the associated free potential energy (see also Hergarten et al., 2014; Savenije, 2017).

In this paper we demonstrate that the linear reservoir behaviour can be seen as the consequence of the catchment organising in such a way that the dissipation of potential energy of the active groundwater by the outflow from the catchment is minimised.

In order to do so, we consider the mass balances of groundwater of two sub-catchments draining towards the outfall of a catchment. In these groundwater bodies we consider the depletion of potential energy of the water as it drains towards the stream. Subsequently we show that the most efficient drainage of the groundwater, in terms of minimising dissipation, leads to a linear reservoir behaviour. We then discuss how this result fits into the larger context of thermodynamic optimisation, and the seemingly counterintuitive hypotheses that state that thermodynamic systems evolve to maximum, rather than minimum,

dissipation states (e.g. Kleidon et al., 2013; Kleidon, 2016).

## 2  Thermodynamics of a groundwater catchment

We consider a catchment that consists of two smaller sub-catchments in terms of their water budget (as illustrated in Fig. 2). We consider these catchments in steady state, and only consider the net precipitation input, $P_i$ (with $i = a, b$ being the index





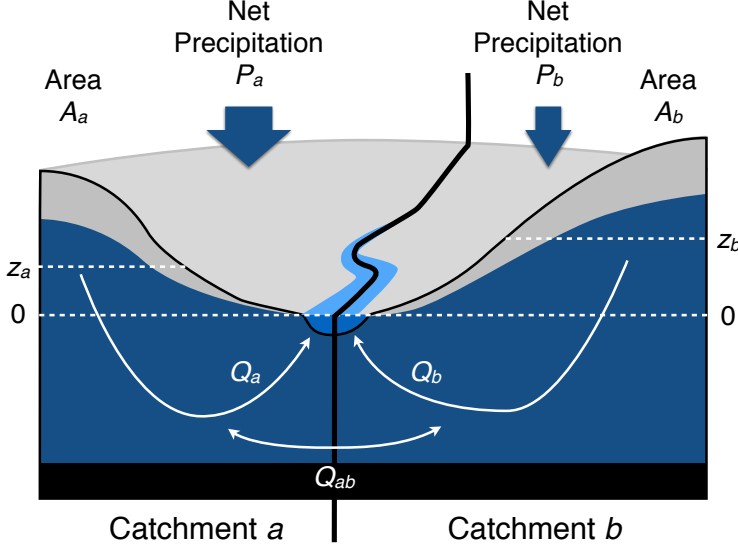

**Figure 2.** Schematic illustration of the idealised catchment formed by two sub-catchments $a$ and $b$ that have areal coverages, $A_a$ and $A_b$.

for the sub-catchments $a$ and $b$), which is the recharge to the phreatic aquifer, as well as the drainage flux from these reservoirs, $Q_i$ for simplicity. The justification for considering the water balances in steady state is that any form of organisation of the drainage structures and the connectivity among the two sub-catchments should take place on a time scale that is much longer than the time scale on which drainage occurs after a wetting event.

In the following, we first formulate the mass balances of these two catchments, which yield expressions for their respective mean groundwater levels, $\bar{z}_a$ and $\bar{z}_b$. The bar over the $z_i$ is to indicate that these are the mean water levels in a steady state mass balance, and not the temporally varying water levels associated with the drainage after a wetting event. The groundwater levels are measured in relation to the water level in the channel, which has a dendritic structure in the catchment and is not a horizontal plain. Also, the reference levels do not have to coincide for the two catchments $a$ and $b$. However, to keep the

following descriptions simple, we set both reference levels to zero. We then describe the expressions for dissipation and then minimise the dissipation associated with the drainage fluxes. This solution is then used to show that the whole catchment in this case acts as a linear reservoir.

    All symbols used are summarised in Table 1.

## 2.1   Mass balances

We first describe the groundwater outflows, $Q_a$ and $Q_b$, of the two sub-catchments as a function of the mean heights of the groundwater levels, $\bar{z}_i$ (equivalently to how it was described in the introduction, except we consider the fluxes here per unit



**Table 1.** Variables used in this study.

| Symbol | Variable | Unit |
|---|---|---|
| $A_a, A_b$ | Areal coverages of the sub-catchments $a$ and $b$ of the whole catchment area | m$^{-2}$ |
| $D_a, D_b, D_{ab}$ | Dissipation rates of potential energy per unit area | W m$^{-2}$ |
| $g$ | Gravitational acceleration | m s$^{-2}$ |
| $n$ | Porosity | % |
| $P_a, P_b$ | Net precipitation (recharge) per unit area | kg m$^{-2}$ s$^{-1}$ |
| $Q_a, Q_b$ | Groundwater outflow per unit area | kg m$^{-2}$ s$^{-1}$ |
| $Q_{ab}$ | Groundwater flow between sub-catchments | kg m$^{-2}$ s$^{-1}$ |
| $\rho$ | Density of groundwater | kg m$^{-3}$ |
| $\tau_a, \tau_b$ | Characteristic time scale | s |
| $\bar{z}_a, \bar{z}_b$ | Mean height of groundwater above channel | m |

area):

$$Q_a = \frac{\bar{z}_a}{\tau_a} \tag{4}$$

and

$$Q_b = \frac{\bar{z}_b}{\tau_b} \tag{5}$$

where the time scales $\tau_i$ describe the effects conductivity and porosity.

We then express the two mass balances of the sub-catchments as

$$A_a \cdot P_a = A_a \cdot \frac{\bar{z}_a}{\tau_a} + A_a \cdot Q_{ab} \tag{6}$$

and

$$A_b \cdot P_b + A_a \cdot Q_{ab} = A_b \cdot \frac{\bar{z}_b}{\tau_b} \tag{7}$$

where $A_a$ and $A_b$ are the areas of the respective sub-catchments. All fluxes in these balance equations are formulated in terms of fluxes per unit area. The flux $Q_{ab}$ describes a connecting redistribution flux between the sub-catchments through the interaction of the groundwater reservoirs within the catchment.

The mass balances can then be used to express the mean water levels of the sub-catchments as functions of precipitation input (the forcing), the time scales (the catchment properties), and the connecting flux between the sub-catchments, $Q_{ab}$, which is so far undetermined. The associated expressions are given by

$$\bar{z}_a = (P_a - Q_{ab}) \cdot \tau_a \tag{8}$$



and

$$\bar{z}_b = \left( P_b + \frac{A_a}{A_b} \cdot Q_{ab} \right) \cdot \tau_b \tag{9}$$

These groundwater levels are linked to the active groundwater storage by $S_i = \rho n_i A_i z_i$, where $\rho$ is the density of water, $n_i$ the porosities, $A_i$ the areas, and $z_i$ the groundwater levels in relation to the water level of the channel network.

## 2.2 Minimum dissipation

In order to evaluate the minimisation of dissipation associated with drainage, we consider the budget of potential energy of the whole catchment. This budget consists of the input of potential energy by precipitation into the groundwater reservoirs at their respective geopotential heights, $g\bar{z}_i$, and the dissipative terms due to drainage from the sub-catchments and due to the flux between sub-catchments. For simplicity, we use a reference level such that the drainage from the sub-catchments to the streams corresponds to a geopotential height of zero. Then, the budget of potential energy input and the dissipative terms is described by

$$A_a \cdot P_a \cdot g\bar{z}_a + A_b \cdot P_b \cdot g\bar{z}_b = D_a + D_b + D_{ab} \tag{10}$$

where $D_i$ are the dissipative terms due to the drainage from the sub-catchments, given by

$$D_a = A_a \cdot Q_a \cdot g\bar{z}_a = A_a \cdot \frac{g\bar{z}_a^2}{\tau_a} \tag{11}$$

and

$$D_b = A_b \cdot Q_b \cdot g\bar{z}_b = A_b \cdot \frac{g\bar{z}_b^2}{\tau_b} \tag{12}$$

and $D_{ab}$ is the dissipation by the redistribution flux, given by

$$D_{ab} = A_a Q_{ab} \cdot g \left( \bar{z}_a - \bar{z}_b \right) \tag{13}$$

We then assume that the minimisation of dissipation associated with drainage is accomplished by the redistribution of groundwater between the sub-catchments. Mathematically, this is achieved by

$$\frac{d(D_a + D_b)}{dQ_{ab}} = \frac{dD_a}{d\bar{z}_a} \cdot \frac{d\bar{z}_a}{dQ_{ab}} + \frac{dD_b}{d\bar{z}_b} \cdot \frac{d\bar{z}_b}{dQ_{ab}} = 0 \tag{14}$$

Using the expressions for $D_a$, $D_b$, and $\bar{z}_a$ and $\bar{z}_b$ from above, the minimisation yields an optimum solution of

$$\bar{z}_a = \bar{z}_b \tag{15}$$

This equality does not mean that there is no head difference between catchments $a$ and $b$. It merely means that they have equal heads in relation to their channel network. In fact, if there was no head difference between the two sub-catchments, then the redistribution flux would be zero. We can obtain an expression for the optimum redistribution flux, $Q_{ab,min}$ (with the subscript





"min" to reflect the outcome of the minimisation of dissipation), by combining Eqs. 8, 9 and 15. Since $\bar{z}_a = \bar{z}_b$, the expression of $Q_{ab,min}$ is obtained by equating Eqs. 8 and 9:

$$Q_{ab,min} = \frac{P_a \cdot \tau_a - P_b \cdot \tau_b}{\tau_a + \frac{A_a}{A_b} \cdot \tau_b} \tag{16}$$

This expression can be understood as the redistribution of the difference in recharge amounts (which would be given by $P_i\tau_i$),
scaled by some effective time scale in the denominator that combines the drainage characteristics of both sub-catchments.

The optimum solution of minimum dissipation results in a mean groundwater level $\bar{z} = \bar{z}_a = \bar{z}_b$ that is obtained by substitution of Eq. 16 in 9:

$$\bar{z} = \frac{A\tau_a\tau_b}{A_b\tau_a + A_a\tau_b} \cdot \frac{(A_aP_a + A_bP_b)}{A} \tag{17}$$

with $A = A_a + A_b$ being the total area of the catchment. This expression describes the mean height in the catchment as a
function of the mean recharge into the catchment, $(A_aP_a + A_bP_b)/A$, at an effective catchment time scale $\tau$ that is obtained by the weighted geometric mean of the time scales of the sub-catchments:

$$\tau = \frac{A\tau_a\tau_b}{A_b\tau_a + A_a\tau_b} = A\left(\frac{A_a}{\tau_a} + \frac{A_b}{\tau_b}\right)^{-1} \tag{18}$$

In other words, the resulting time scale that results from the optimisation is such as if the two flow resistances of the groundwater reservoirs of the sub-catchments act as if they were in linked parallel in an electric circuit (see also the same expression
in Savenije, 2017).

The drainage of the whole catchment can then be described by

$$Q = Q_a + Q_b = A \cdot \frac{\bar{z}}{\tau} \tag{19}$$

so that the whole catchment acts as a linear reservoir with an effective time scale given by Eq. 18.

## 3 Discussion

Our treatment of groundwater flow here is, of course, highly simplified, and is thus potentially impacted by some limitations. What we show here is that the exponential recession of streamflow of larger catchments can be understood by the connectivity of the smaller-scale sub-catchments. The exponential recession can, of course, also be caused by other factors. For instance, sub-catchments within the region can evolve at similar rates. Similar rates of weathering may then result in similar dynamics that may lead to comparable time scales $\tau$. When the sub-catchments drain at similar time scales, these could also result in an
approximately linear reservoir behaviour at the larger scale. Also, as some authors have pointed out, not all river basins operate as linear reservoirs (Wittenberg, 2003; Moore, 1997; Chapman, 1999). On the other hand, our approach here not just shows that the linear reservoir can result from connectivity and organisation of the groundwater flow, it also makes a prediction of the effective time scale of discharge of the catchment. When we use the example of the Ourthe river from the introduction, we find that by using Eq. 18 with the time scales of the two branches and their respective areas (Ourthe Occidentale with $\tau_a = 1660$ h





and $A_a = 379$ km$^2$; Ourthe Orientale with $\tau_b = 1910$ h and $A_b = 317$ km$^2$), this yields a value of $\tau = 1765$ h, which is very close to the derived $\tau = 1772$ h from Fig. 1. While this is certainly no rigorous test of our interpretation, it nevertheless points out that it results in a reasonable prediction of the behaviour of the whole catchment and would thus appear to be a plausible explanation.

We can then ask how catchments would be able to organise in such a way that the dissipation associated with groundwater outflow is minimised, and what the likely conditions are for such an emergent outcome. The outcome that the mean groundwater levels of the two sub-catchments in relation to their channel network are equal that resulted from the minimisation suggests that the sub-catchments are so well connected that the balancing of the mean groundwater levels between them acts at a faster time scale than the overall groundwater outflow from the catchment. This suggests that this behaviour requires a sufficiently evolved

stage of catchment evolution in which there was sufficient time to connect sub-catchments by some fractal-like networks, as e.g., proposed by Hergarten et al. (2014) and Savenije (2017). This interpretation would imply that the linear reservoir behaviour is probably less likely to be observed in young catchments, in arid regions as groundwater reservoirs may not be connected, and in mountainous catchments where steep slopes result in a shorter timescale for groundwater outflow. Evaluating this aspect further could form an interesting topic for future research.

At a more general level, our work raises questions as to which hydrologic flux can be inferred from the application of a thermodynamic optimality approach, whether this is by minimisation or maximisation of dissipation (which appear rather contradictory), and whether there are more general insights that can be gained from our work. To do so, we first briefly describe applications of thermodynamic optimality, specifically maximum power (or entropy production), in the atmospheric sciences as there is a substantial body of literature dealing with such applications (see e.g., review by Ozawa et al., 2003). We then use

this application to identify the similarities and differences in the processes and interactions with boundary conditions that result in the form of thermodynamic optimality.

    The application of thermodynamic optimality to constrain atmospheric motion is based on the two principal means by which energy can be transported, either by radiation (in form of radiant energy) or by motion (in form of heat). The emission of radiation as well as heat transport by motion deplete temperature differences, but the rate by which motion depletes this

difference depends on the power involved to drive the flow. With little motion, heat transport contributes little to the depletion of temperature differences, and radiation accomplishes the greater part. With increasing levels of motion, more of the depletion is accomplished by heat transport, but the temperature difference is further reduced. As power is proportional to the product of the heat flux and temperature difference and power balances dissipation in steady state, such a state is equivalent to maximum dissipation and, approximately, a maximum in entropy production. Yet, the example of groundwater and the linear reservoir

described here is different, because there are no two different types of processes that "compete" to deplete the potential energy of the groundwater levels. This is an important difference in the setup of the groundwater system compared to applications of thermodynamic optimality in atmospheric sciences.

    The setup we consider here does not represent a "competition" between two kinds of fluxes, as the water balances of the groundwater reservoirs as we formulated them here can only drain by the mass fluxes associated with groundwater outflow. The

mass balance in steady state requires that the input by recharge is balanced by the outflow, which imposes a strong constraint





on the mass balance. Our setup nevertheless allows for a competition of the outflow directly from the respective reservoirs, or through the connecting flux $Q_{ab}$. This flux was used here to minimise the total dissipation associated with the groundwater outflow. In principle, we could have also asked whether this flux, $Q_{ab}$, maximises the dissipation $D_{ab}$. A greater flux $Q_{ab}$ results in a lower difference in the groundwater levels between the two sub-catchments, so that a state of maximum dissipation

exists associated with this flux. It results in an optimum flux $Q_{ab,max} = Q_{ab,min}/2$, which results in a different outcome where the groundwater levels are not equalised ($\bar{z}_a \neq \bar{z}_b$). Also, we could have asked why the time scales $\tau_a$ or $\tau_b$ could not be derived by optimisation. A smaller value of $\tau_i$ would imply less resistance to drainage, yet it also results in a lower value of the groundwater level $\bar{z}_i$, so that dissipation associated with drainage decreases with smaller $\tau_i$. What this points out is that the level of dissipation associated with the process of groundwater outflow in steady state does not only depend on the frictional loss (as

represented by the time scale $\tau_i$), but also on the interaction with the boundary condition, which is set by the groundwater level $z_i$ and which sets the magnitude of how much potential energy is being dissipated by the system. The way in which we set up the system that we consider thus plays quite a critical role in how to apply thermodynamic optimality.

    The issue about which process to optimise by maximisation or minimisation becomes clearer when we consider groundwater outflow in a broader systems context in which we consider the interface in which water enters the soil and the fluxes that affect

the overall recharge of groundwater. When we draw our system boundary at the top of the soil column, then the input of potential energy by recharge becomes essentially independent of the groundwater drainage flux because the elevation of the top of the soil column is set by processes of uplift and erosion and thus by processes of a different kind, and not by groundwater drainage. The net input of potential energy into this broader system is then fixed, so that the rates at which work is performed and this potential energy is dissipated is fixed as well. A minimisation of dissipation by groundwater outflow then implies that a

maximum of work can be derived from the input of potential energy at the top of the soil column by other processes. This work that can be used, for instance, to build and maintain preferential flow networks in the unsaturated zone, processes that allow for a faster drainage of precipitation input by subsurface runoff to the river outlet. It would thus seem that the minimum dissipation of groundwater outflow would correspond to a maximum power and dissipation associated with fast drainage of the unsaturated zone, which agrees with Rodriguez-Iturbe and Rinaldo (1997) assuming minimum energy expenditure in the river networks and

with Kleidon et al. (2013) where minimum energy expenditure (or dissipation) in the river system leads to maximum power of erosion outside the river network. While this aspect would clearly need to be looked at in greater detail, it nevertheless emphasises the importance on a system's perspective that includes a broader range of dissipative processes involved in river discharge, and a more differentiated view on which processes dominate structure formation and the dissipative activity of the whole catchment.

## 30    4   Summary and conclusions

We used a simple setup of two sub-catchments to show that when these are connected, their joint response also results in a linear reservoir when the overall dissipation associated with groundwater outflow is minimised. The minimisation of dissipation was achieved by a groundwater flux that connects the two sub-catchments. We interpreted this to imply that the behaviour of



large catchments can be described by a linear reservoir in regions in which the groundwater reservoirs of sub-catchments are sufficiently well connected and evolved. It presents a reasonable and parsimonious explanation for why an exponential recession can be observed in the river flow of larger-scale catchments, and provides some criteria for when such a response can be expected to be observed. It would thus seem that thermodynamic optimality can provide an additional constraint to reduce

equifinality in catchment behaviour.

We also argued that a broader system perspective is important when applying thermodynamic optimality approaches to hydrologic fluxes, specifically regarding which flux to optimise, and whether dissipation is minimised or maximised. In the example we described here, the groundwater levels described the input of potential energy into the system, so that the minimisation of dissipation affected this input of energy. When the broader system context of water input into the soil is used,

then this energy input is fixed so that the minimisation of dissipation with groundwater outflow would be intimately linked to a maximisation of dissipation of drainage processes in the unsaturated zone. While this aspect would need to be further evaluated and specified in future work, it nevertheless emphasises the need for an encompassing systems' perspective that is based on thermodynamics to explain the overall organisation and functioning of catchments and their evolution.

*Acknowledgements.* AK acknowledges helpful discussions with Anke Hildebrandt and support by the German Science Foundation (DFG)

through the "Catchments As Organized Systems (CAOS)" research group.



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
