# Peer review of "Minimum dissipation of potential energy by groundwater outflow results in a simple linear catchment reservoir"

_Hydrology and Earth System Sciences, 2017_

## Referee Comment (RC1) · M.C. Westhoff (Referee) · 28 Nov 2017

Review of "Minimum dissipation of potential energy by groundwater outflow results in a simple linear catchment reservoir" - Kleidon and Savenije.

In this manuscript, the authors show that simple linear behaviour of a catchment results from the minimization of dissipation by the draining fluxes of two sub-catchments, in which a flux between the two catchments optimizes itself to reach this minimum. To my opinion, it is a very novel contribution, shedding a new light on why so many catchments behave as a linear reservoir. However, there are some minor issues which should be

improved before publication.

The first one is about the main message: The authors claim that the linear behaviour is caused by a connecting flux between two sub-catchments, which minimizes the dissipation of the draining fluxes. However, they start with the assumption that the sub-catchments already behave as a linear reservoir. So what this manuscript tells us that if two sub-catchments behave linearly, the sum of the two also reacts as a linear reservoir. And with the proposed method it is possible to also predict the timescale and mean height above the drain. It does NOT explain why a sub-catchment behaves like a linear reservoir.

A second point considers the used units: Especially the units of the fluxes are inconsistent throughout the manuscript. In Eq. 1-9 and 17-19 they are given in m/s, while in Eq. 10-14 and table 1 they are given as a mass flux per unit area. Also the storage S is not consistent: in Eq. 1 it is in m, while directly after Eq. 9 it is given in kg. The dissipation terms (D) in table 1 are given in W/m2, while in Eq. 10-13 they should be in Watts.

A third point is that in the example of the Ourthe, the authors show that tau can be reasonably well predicted. However, they do not report other state variables or fluxes. For example, when a precipitation of 1000 mm/year is assumed, $z_a$ and $z_b$ would be about 20 cm. The authors should make it plausible that this is a reasonable value.

As a last point I think the authors take to big steps when discussing why the dissipation by $Q_{ab}$ should not be maximised (P9, L3-6): The main reason they give is that when doing this, $z_a$ is not equal to $z_b$. But why do they have to be the same? The only reason I can think of is that with an equal $z$, it is possible to obtain an effective tau, which is close to (only) one observation. But one observation does not prove the hypothesis of this manuscript. The authors do recognize this 'lack' of sufficient observations to fully accept their hypothesis, but they are, in my opinion, too fast in rejecting the hypothesis of maximum dissipation by $Q_{ab}$.

Some very minor points:

P1, L4: maybe refer to the master recession curve as well? Or maybe even better: create a Master recession curve of the Ourthe data.

P3, L9: from the analysis in this paper, leakage is even required to obtain a linear reservoir.

P4, L10: In the derivations, it does not seem to matter if the reference level is set to zero or not. However, I suggest to leave arbitrary reference levels in the derivation, because this simplification causes $Q_{ab}$ to be zero in the optimum state, while the point the authors want to make is that this flux is essential to obtain this optimum state

Table 1: $A_a$ and $A_b$ are in m$^2$

Eq. 11: Add that dissipation is determined under the assumption that the change in kinetic energy is negligible.

Eq. 18: Although this equation gives a weighted mean, it is not the geometric mean, which is given by $\tau_a^{A_a/A} * \tau_b^{A_b/A}$.

With kind regards,

Martijn Westhoff

---

## Short Comment (SC1) · 2 Dec 2017

This study clarifies a catchment's overall behavior by deriving a characteristic time-scale for steady-state conditions. It seems to me that any catchment (i.e. also a non-linear one) in a single state (e.g. steady-state) can be characterized by a single characteristic time-scale (tau = S/Q). However, a catchment can only be considered to behave as a linear reservoir when this characteristic timescale tau also applies in other states (e.g. lower or higher storage and runoff). For the presented example, showing that the derived timescale tau also applies to other conditions seems to require relaxing the steady-state assumption. However, in that case, the characteristic timescale cannot be

derived anymore with the presented analysis. Thus, do we now have a Catch-22? Or do I misunderstand something?

(P.s. a short conversation with the second author could not clarify this issue and he encouraged me to post this query on HESSD)

Kind regards,

Wouter Berghuijs

---

## Short Comment (SC2) · 4 Dec 2017

Comment on **"Minimum dissipation of potential energy by groundwater outflow results in a simple linear catchment reservoir"** by A. Kleidon and H.H.G. Savenije

**Summary of the Paper**

Kleidon and Savenije give an explanation why the simple linear reservoir can often be used as a model to describe groundwater outflow to a stream, despite the very complex nature of real catchments. The authors show that the linear reservoir equation can be derived based on the assumption that the potential energy of groundwater of a whole catchment dissipates at minimum rate. They derive the model analytically for idealised conditions. They then discuss why this concept was chosen and how it fits into broader (thermodynamic systems) theory.

**Comments**

Having read the review of Martijn Westhoff, I realise that I repeat some issues that were already mentioned. However, I didn't remove them since this repetition might suggest that they are actually important.

Generally, the paper is well written and an interesting and innovative contribution to catchment hydrology. It is an ongoing research question, whether and why we so often observe a linear behaviour at the catchment scale. On the contrary, non-linear behaviour is frequently observed, too, and different concepts exist to explain that (e.g. Brutsaert and Nieber, 1977; Wittenberg, 1999; Harman et al., 2009). Although literature on thermodynamic optimality in hydrology exists, the employed approach is rather unusual and based on many assumptions (see comments below).

**Sub-catchments** The authors use the linear reservoir equation also for the sub-catchments they start with (see Eq. 4: $Q_a = \frac{\bar{z}_a}{\tau_a}$ and Eq. 5). As pointed out earlier and also stated in the paper, we often observe non-linear behaviour (the authors mention that the linear reservoir is "often contested [...] particularly by authors who look at processes at smaller scales"). It would be interesting to read a discussion whether this assumption limits the validity of the result. What would happen if one started with non-linear sub-catchments?

Generally, the authors ask the question why catchments behave as a simple linear reservoir despite the complex reality. It is perfectly reasonable to start with a simple example. However, a result using two very simplified sub-catchments is perhaps not an answer to "why does such a simple equation pertain, while [...] the subsurface is extremely heterogeneous". What if we had more than two sub-catchments? What would heterogeneity result in (e.g. in effective rainfall, soil properties, ...)?

**Flux between sub-catchments** The authors assume that the redistribution flux $Q_{ab}$ is happening on a faster time scale than the groundwater outflow. It would be interesting to read whether there is more evidence for that assumption (perhaps observations)?

**(Sub-)Catchment geometry** If I understand it correctly, the example shows two sub-catchments that drain to a specific sub-stream, which then discharges to the large stream. This setup is different to the one shown on Figure 2, where it seems as if both sub-catchments drain to the same stream. The authors mention on page 4, line 8, that the channel has a dendritic structure. After reading the whole paper, I suppose this points to cases like in the example, where there are two (or perhaps more) channels eventually draining to one channel and not just one central channel throughout the whole catchment. Maybe it is just me not understanding it correctly. However, it might be helpful if the setting would be explained more clearly, and also to know if the example is comparable in terms of its general geometry.

**Ourthe river example** The authors clearly state that their quick calculation is no rigorous validation and this might not be the emphasis of this work. Nonetheless, could it be made more rigorous by adding some more calculations? This could imply other data from the same catchments to see if they are consistent, and data from other similarly arranged (sub-)catchments. The current time-scales are all quite similar, and probably the associated soil and hydro(geo)logical properties. This means that the resulting value of an overall time constant lying between the two sub-catchment time constants could perhaps be explained otherwise, e.g. by simple averaging (which I assume is implied by page 7, line 24). Such further calculations could probably be done quite quickly and would certainly give more insight.

**Justification of minimum dissipation approach** The minimisation of dissipation by groundwater outflow is the main hypothesis and it is underpinned by looking at a broader system, including the unsaturated zone. The authors hypothesise that dissipation is minimised in the saturated zone, while being maximised in the unsaturated zone. This theory is strengthened by referring to similar approaches dealing with river systems. Is there any other evidence for the minimum dissipation approach (in preference to the maximum dissipation approach)? Furthermore, can the separation between saturated/unsaturated zone be assumed to be that distinct in nature, considering e.g. a varying groundwater table or a large capillary fringe. Although it is admitted, that many of the hypotheses need to be tested and looked at in more detail, I think further arguments (or empirical evidence, as mentioned above) would help to strengthen the main hypothesis.

**Units** There are some inconsistencies regarding the units. In Table 1, the unit of area is given as $m^{-2}$, but should rather be $m^2$. Also, according to Table 1, $Q_i$, $P_i$, etc., are mass fluxes. The transformation to volume fluxes is straightforward if we assume that density and porosity are constant (this transformation is indicated on page 6, line 4). However, many equations miss the density $\rho$ and porosity $n$ if the $Q_i$ are assumed to be mass fluxes (e.g. $Q_i = \frac{\bar{z}_i}{\tau_i}\rho n_i$). It seems to cancel out in the following equations, so that it should not influence the resulting equations.

**Summary**

The paper tries to explain an interesting and still unresolved question in an innovative way and is hence clearly of value for the hydrological sciences. It contains many assumptions, most of them simplifying the setup. While this is reasonable in order to find a simple analytical solution, their impact could be discussed more extensively (what if they are not satisfied?). The main assumption is the minimisation of dissipation. Since it forms the basis of the whole approach, a more detailed reasoning and discussion of that would be desirable. The paper focuses on the presentation of an idea and not on its validation, which is generally fine. However, more testing (calculations) of the proposed theory should be relatively straightforward and could justify the theory in an empirical way.

Kind regards,

Sebastian Gnann

**References**

Brutsaert, W. and Nieber, J. L. (1977). Regionalized drought flow hydrographs from a mature glaciated plateau. *Water Resour. Res*, 13(3):637–643.

Harman, C., Sivapalan, M., and Kumar, P. (2009). Power law catchment-scale recessions arising from heterogeneous linear small-scale dynamics. *Water Resources Research*, 45(9).

Wittenberg, H. (1999). Baseflow recession and recharge as nonlinear storage processes. *Hydrological Processes*, 13(5):715–726.

---

## Author Comment (AC1) · 5 Dec 2017

Wouter Berghuijs raised an important point about the steady-state assumption in our manuscript. He described that the linear reservoir describes behavior of catchments that are not in steady state and hence asked how our result using the steady-state assumption would apply.

In the manuscript, we used the steady state assumption because we argued that the development of connecting structures would take place over time scales much longer than the discharge after a wetting event. The time scale that we derived from the optimization in steady state then also applies to the discharge on time scales of the

discharge after a wetting event. This is because the time scale that characterizes the exponential recession curve and the mean discharge over the same time period is the same. This can be seen as follows:

The drainage after a single wetting event is described by (Eq. 2 in the manuscript)

$$Q(S) = \frac{S}{\tau} \tag{1}$$

where $Q(S)$ is the discharge (which is a function of $S$), $S(t)$ is the active storage of groundwater (which is a function of time $t$), and $\tau$ is the characteristic time scale. The exponential recession curve is then described by

$$Q = Q_0 e^{-t/\tau} \tag{2}$$

where $Q_0$ is the discharge at time $t = 0$.

When averaged over a time interval $\Delta t$, the mean drainage $Q_{mean}$ during this time is described by:

$$Q_{mean} = \frac{1}{\Delta t} \int_0^{\Delta t} Q_0 e^{-t/\tau} dt = \frac{1}{\tau} \frac{Q_0 (1 - e^{-\Delta t/\tau})}{\Delta t} \tag{3}$$

Active storage $S$ varies with time in a similar fashion, and is described by

$$S = Q(t)\tau \tag{4}$$

When averaged over the same time interval $\Delta t$, the mean value of active storage $S_{mean}$ is described by

$$S_{mean} = \frac{1}{\Delta t} \int_0^{\Delta t} S_0 e^{-t/\tau} dt = \frac{Q_0 (1 - e^{-\Delta t/\tau})}{\Delta t} \tag{5}$$

where $S_0 = Q_0 \tau$ is the active storage at time $t = 0$. The time scale derived from the ratio of mean active storage to mean discharge then yields the same time scale as in

the linear reservoir:

$$\tau_{mean} = \frac{S_{mean}}{Q_{mean}} = \tau \qquad (6)$$

In other words, the time scales that characterize the mean behavior and the instantaneous behavior in the linear reservoir are identical.

Hence, the time scale that we derived from the steady-state assumption by optimization describes the transient behavior after a wetting event as well.

We will clarify this point in the revision of the manuscript.
* * *

---

## Short Comment (SC3) · 9 Dec 2017

The work by the authors is quite interesting. However, the main premise that catchments behave as linear reservoirs needs to be properly reinforced.

It is claimed that large catchments behave as linear reservoirs although hill-slopes and small catchments behave as non-linear reservoirs (line 5). However, many recent studies have shown that large catchments generally behave as non-linear system (e.g., Biswal and Marani, 2010; Shaw and Riha, 2012; Mutzner et al., 2013; Biswal and Nagesh Kumar, 2014; Ben Krewajski, 2016). At least a through discussion on this subject is needed.

[Figure]

Cannot the concept of "minimum dissipation of potential energy" explain catchment non-linearity?

References:

Biswal, B., & Marani, M. (2010). Geomorphological origin of recession curves. Geophysical Research Letters, 37(24).

Biswal, B., & Kumar, D. N. (2014). What mainly controls recession flows in river basins?. Advances in Water Resources, 65, 25-33.

Chen, B., & Krajewski, W. (2016). Analysing individual recession events: sensitivity of parameter determination to the calculation procedure. Hydrological Sciences Journal, 61(16), 2887-2901.

Mutzner, R., Bertuzzo, E., Tarolli, P., Weijs, S. V., Nicotina, L., Ceola, S., ... & Rinaldo, A. (2013). Geomorphic signatures on Brutsaert base flow recession analysis. Water Resources Research, 49(9), 5462-5472.

Shaw, S. B., & Riha, S. J. (2012). Examining individual recession events instead of a data cloud: Using a modified interpretation of dQ/dt–Q streamflow recession in glaciated watersheds to better inform models of low flow. Journal of hydrology, 434, 46-54.

---

## Referee Comment (RC2) · S. Hergarten (Referee) · 15 Dec 2017

The manuscript by Axel Kleidon and Hubert H. G. Savenije presents a concept for explaining why catchments could behave like linear reservoirs. The theory assumes two coupled reservoirs draining into one catchment and consists of two parts. First it is shown that it is energetically favorable if both reservoirs are on the same groundwater level with respect to their river. Then it is shown that they behave like a single linear reservoir under this condition, and the resulting characteristic time scale is computed.

While I find the overall concept interesting and enjoyed reading the manuscript, I found some critical aspects among which one might even question the concept as a whole. In

principle, my concerns addressed in the following have already been mentioned by the contributors to the discussion, Martijn Westhoff, Wouter Berghuijs, Sebastian Gnann, and Basudev Biswal.

**Linearity of the catchment:** There has been discussion to what extent real catchments behave like linear reservoirs. This point has also been pointed out by Basudev Biswal. I share the opinion that linear behavior is not a universal property, at least not at all time scales. There should indeed be more discussion on this aspect. However, even if I am personally not convinced of linear catchment behavior, this aspect does not question the merit of this study in my opinion. Explaining why some catchments are more linear than one might expect would also be valuable.

**Linearity of the two reservoirs:** As already pointed out by Martijn Westhoff, the authors assume that both reservoirs are already linear, which should be clarified.

**Energy dissipation of the flow between the two reservoirs:** This point was already raised by Martijn Westhoff. They authors neglect the energy consumed for the flow between the two coupled reservoirs. I guess that the total energy dissipation would be constant if this component was taken into account, so that there would be nothing left to minimize. The explanation given in the paper is indeed somewhat weak. On the other hand, models involving optimization often minimize only some components where the reasons are sometimes not straightforward. So this point would also require more explanation, but would not be crucial in my opinion.

**Time scale of the flow between the two reservoirs:** The authors assume that the optimization holds for all times, so that the two reservoirs are at the same groundwater level at each time. This implies that flow between the reservoirs takes place instantaneously and without any limitation. In other words, the time scale of this flow must be zero or at least be much smaller than the time scale of the faster of the two reservoirs. This point was raised by Sebastian Gnann, and the question about stationarity asked by Wouter Berghuijs goes into the same direction.

This assumption appears to be unrealistic to me. What could be a structure of two reservoirs where the communication between both is much faster than their drainage to the river? If we release this restriction, i.e., if we only assume that the two reservoirs always want to be in an optimized state, but adjustment involves a finite time scale $\tau_{ab}$, the entire result would be lost. We would then get stuck at a superposition of two exponential recession curves where the two time scales are determined by the eigenvalues of the matrix

$$A = \begin{pmatrix} \frac{1}{\tau_a} + \frac{1}{\tau_{ab}} & -\frac{1}{\tau_{ab}} \\ -\frac{1}{\tau_{ab}} & \frac{1}{\tau_b} + \frac{1}{\tau_{ab}} \end{pmatrix} .$$

Then the optimization would not change anything on the behavior that two linear reservoirs yield a superposition of two exponential recession functions, only with different characteristic time scales.

And if we accept that this time scale must be zero, so that the reservoirs are always at the same groundwater level? Would we still consider them as two distinct reservoirs? Or is it not rather a single reservoir then? In this case, the results would not be very surprising.

The last aspect seems to be critical to me as it might challenge the whole concept. I hope that there will be more contributions to the discussion about this point as I may also be wrong with this concern.

Best regards,

Stefan Hergarten

---

## Referee Comment (RC3) · S. Hergarten (Referee) · 21 Dec 2017

Dear authors;

Thanks for your prompt reply! I fully agree that all points except for the last one can easily be fixed with some more explanation. However, I am still not convinced about the last point. There was indeed a little misunderstanding, but I am afraid that the clarification does not solve the problem.

When reading the paper I was happy with the consideration of mean values until I arrived at the last sentence of Sect. 2. There the main conclusion of the paper is drawn

from Eq. (19) – the linear behavior of the reservoir as a whole. However, Eq. (19) in its pure form with the mean value definitely does not allow for this conclusion. It only describes the slope at one point of the recession curve which is not much information. Linear behavior would require that Eq. (19) must hold for all times. Otherwise the coupled reservoir would still be described by the superposition of two exponential recession curves as pointed out in my review. As this seemed to be so obvious to me I thought that you implicitly generalized Eq. (19) from the mean value to all times in order to be able to draw this conclusion, and that you just found it not necessary to explain this in detail. So after realizing now that all relations should indeed be restricted to the mean value, the problem has just been shifted. Then the conclusion of linear behavior is not correct in my opinion.

However, I would be very happy if someone else, perhaps one of the other contributors to the discussion, could take a look at this point. I may be wrong, but if not, this would somehow make a major part of the paper collapse.

---

## Author Comment (AC2) · 21 Dec 2017

We would like to thank the reviewer for his comments. While the first comments he made are relatively minor points that we will respond to in the final response, we want to respond to the last point as he considers this to be a major point. We think this last point is based on a misunderstanding which may be caused by our inadequate explanation in the text.

The last point is about the time scale of flow between the two reservoirs and of the optimization. We want to clarify first that we do **not** assume that the optimization holds all the time, but that it holds in the **mean**. This means, we do not enforce optimality at

each step of the streamflow recession curve, but rather apply the optimization to the mean discharge (we refer to this mean state in section 2, where we also refer to mean groundwater levels). The justification for doing the optimization on the mean is that the processes involved in the optimization likely involve the formation of dendritic flow networks in the groundwater, which takes place on much longer time scales than the time scale of a streamflow recession event. That we find the mean (relative) groundwater levels to be the same as the outcome of the optimization does not imply that the instantaneous groundwater levels are the same, and it also does not imply that the flow between the catchments is instantaneous.

On this aspect we would also like to point out that it actually requires very little flow between the catchments to accomplish the outcome of the optimization. One can use the equations from the manuscript to see that if recharge is about the same for both catchments, the flow between them is $Q_{ab}/Q = (\tau_a - \tau_b)/(2(\tau_a + \tau_b))$, which is likely to be much smaller than one. For the example of the Ourthe catchment, this amounts to about 3.5% of the discharge, which is a comparatively small flux. This small flux implies a much longer time scale than the one involved in the stream flow recession, which is consistent with the assumption that the optimization takes place on a longer time scale.

So we do not think that the interpretation by the reviewer is correct, but that this aspect would clearly need to be better described in the manuscript.

---

## Author Comment (AC3) · 15 Feb 2018

In his last comment from 21 Dec 2017, the reviewer raises a point regarding the use of means in our paper. Specifically, he argues that the derivation of the effective mean time scale (Eq. 19) does not allow for the conclusion that the whole catchment acts as a linear reservoir, as it only represents the mean behavior of the catchment.

We agree with the point that we only deal with the mean behavior in the optimization, and that we did not show explicitly that the mean effective time scale implies an exponential recession curve and that the whole catchment acts as a linear reservoir. However, as we explained in our first response to Wouter Berghuijs, for the linear reservoir

the time scales that characterize the exponential recession curve and that characterizes the mean discharge are identical. It would thus seem reasonable to expect that the derivation of the effective mean time scale (Eq. 19) would also be associated with an exponential recession associated with this mean time scale. However, we agree that we do not explicitly show this in the manuscript.

In the revision, we will include an example to illustrate the connection between the mean, optimized behavior and the dynamics of a recession event to address this point and describe this link more explicitly.